# Augur: Data-Parallel Probabilistic Modeling

**Jean-Baptiste Tristan**[1]**, Daniel Huang**[2]**, Joseph Tassarotti**[3]**,**
**Adam Pocock**[1]**, Stephen J. Green**[1]**, Guy L. Steele, Jr**[1]
[1]Oracle Labs {`jean.baptiste.tristan, adam.pocock,`
`stephen.x.green, guy.steele`}`@oracle.com`
[2]Harvard University `dehuang@fas.harvard.edu`
[3]Carnegie Mellon University `jtassaro@cs.cmu.edu`

## Abstract

Implementing inference procedures for each new probabilistic model is time-consuming and error-prone. Probabilistic programming addresses this problem by allowing a user to specify the model and then automatically generating the inference procedure. To make this practical it is important to generate high performance inference code. In turn, on modern architectures, high performance requires parallel execution. In this paper we present Augur, a probabilistic modeling language and compiler for Bayesian networks designed to make effective use of data-parallel architectures such as GPUs. We show that the compiler can generate data-parallel inference code scalable to thousands of GPU cores by making use of the conditional independence relationships in the Bayesian network.

## 1 Introduction

Machine learning, and especially probabilistic modeling, can be difficult to apply. A user needs to not only design the model, but also implement an efficient inference procedure. There are many different inference algorithms, many of which are conceptually complicated and difficult to implement at scale. This complexity makes it difficult to design and test new models, or to compare inference algorithms. Therefore any effort to simplify the use of probabilistic models is useful.

Probabilistic programming [1], as introduced by BUGS [2], is a way to simplify the application of machine learning based on Bayesian inference. It allows a separation of concerns: the user specifies *what* needs to be learned by describing a probabilistic model, while the runtime automatically generates the *how*, i.e., the inference procedure. Specifically the programmer writes code describing a probability distribution, and the runtime automatically generates an inference algorithm which samples from the distribution. Inference itself is a computationally intensive and challenging problem. As a result, developing inference algorithms is an active area of research. These include deterministic approximations (such as variational methods) and Monte Carlo approximations (such as MCMC algorithms). The problem is that most of these algorithms are conceptually complicated, and it is not clear, especially to non-experts, which one would work best for a given model.

In this paper we present *Augur*, a probabilistic modeling system, embedded in Scala, whose design is guided by two observations. The first is that if we wish to benefit from advances in hardware we *must* focus on producing highly *parallel* inference algorithms. We show that many MCMC inference algorithms are highly data-parallel [3, 4] *within a single Markov Chain*, if we take advantage of the conditional independence relationships of the input model (e.g., the assumption of i.i.d. data makes the likelihood independent across data points). Moreover, we can *automatically* generate good data-parallel inference with a compiler. This inference runs efficiently on common highly parallel architectures such as Graphics Processing Units (GPUs). We note that parallelism brings interesting trade-offs to MCMC performance as some inference techniques generate less parallelism and thus scale poorly.

The second observation is that a high performance system begins by selecting an appropriate inference algorithm, and this choice is often the hardest problem. For example, if our system only implements Metropolis-Hastings inference, there are models for which our system will be of no use, even given large amounts of computational power. We must design the system so that we can include the latest research on inference while reusing pre-existing analyses and optimizations. Consequently, we use an intermediate representation (IR) for probability distributions that serves as a target for modeling languages and as a basis for inference algorithms, allowing us to easily extend the system. We will show this IR is key to scaling the system to very large networks.

We present two main results: first, some inference algorithms are highly data-parallel and a compiler can *automatically* generate effective GPU implementations; second, it is important to use a symbolic representation of a distribution rather than explicitly constructing a graphical model in memory, allowing the system to scale to much larger models (such as LDA).

## 2 The Augur Language

We present two example model specifications in Augur, latent Dirichlet allocation (LDA) [5], and a multivariate linear regression model. The supplementary material shows how to generate samples from the models, and how to use them for prediction. It also contains six more example probabilistic models in Augur: polynomial regression, logistic regression, a categorical mixture model, a Gaussian Mixture Model (GMM), a Naive Bayes Classifier, and a Hidden Markov Model (HMM). Our language is similar in form to BUGS [2] and Stan [6], except our statements are *implicitly parallel*.

### 2.1 Specifying a Model

The LDA model specification is shown in Figure 1a. The probability distribution is a Scala object (`object LDA`) composed of two declarations. First, we declare the support of the probability distribution as a class named `sig`. The support of the LDA model is composed of four arrays, one each for the distribution of topics per document (`theta`), the distribution of words per topic (`phi`), the topics assignments (`z`), and the words in the corpus (`w`). The support is used to store the inferred model parameters. These last two arrays are flat representations of ragged arrays, and thus we do not require the documents to be of equal length. The second declaration specifies the probabilistic model for LDA in our embedded domain specific language (DSL) for Bayesian networks. The DSL is marked by the `bayes` keyword and delimited by the enclosing brackets. The model first declares the parameters of the model: `K` for the number of topics, `V` for the vocabulary size, `M` for the number of documents, and `N` for the array of document sizes. In the model itself, we define the hyperparameters (values `alpha` and `beta`) for the Dirichlet distributions and sample `K` Dirichlets of dimension `V` for the distribution of words per topic (`phi`) and `M` Dirichlets of dimension `K` for the distribution of topics per document (`theta`). Then, for each word in each document, we draw a topic `z` from `theta`, and finally a word from `phi` conditioned on the topic we drew for `z`.

The regression model in Figure 1b is defined in the same way using similar language features. In this example the support comprises the (`x`, `y`) data points, the weights `w`, the bias `b`, and the noise `tau`. The model uses an additional `sum` function to sum across the feature vector.

### 2.2 Using a Model

Once a model is specified, it can be used as any other Scala object by writing standard Scala code. For instance, one may want to use the LDA model with a training corpus to learn a distribution of words per topic and then use it to learn the per-document topic distribution of a test corpus. In the supplementary material we provide a code sample which shows how to use an Augur model for such a task. Each Augur model forms a distribution, and the runtime system generates a `Dist` interface which provides two methods: `map`, which implements maximum a posteriori estimation, and `sample`, which returns a sequence of samples. Both of these calls require a similar set of arguments: a list of additional variables to be observed (e.g., to fix the `phi` values at test time in LDA), the model hyperparameters, the initial state of the model support, the model support that stores the inferred parameters, the number of MCMC samples and the chosen inference method.

```
1  object LDA {
2  class sig(var phi: Array[Double],
3          var theta: Array[Double],
4          var z: Array[Int],
5          var w: Array[Int])
6  val model = bayes {
7  (K:Int,V:Int,M:Int,N:Array[Int]) => {
8    val alpha = vector(K,0.1)
9    val beta = vector(V,0.1)
10   val phi = Dirichlet(V,beta).sample(K)
11   val theta = Dirichlet(K,alpha).sample(M)
12   val w =
13     for(i <- 1 to M) yield {
14       for(j <- 1 to N(i)) yield {
15         val z: Int =
16           Categorical(K,theta(i)).sample()
17         Categorical(V,phi(z)).sample()
18       }}
19   observe(w)
20 }}}
```

```
1  object LinearRegression {
2  class sig(var w: Array[Double],
3          var b: Double,
4          var tau: Double,
5          var x: Array[Double],
6          var y: Array[Double])
7  val model = bayes {
8   (K:Int,N:Int,l:Double,u:Double) => {
9    val w = Gaussian(0,10).sample(K)
10   val b = Gaussian(0,10).sample()
11   val tau = InverseGamma(3.0,1.0).sample()
12   val x = for(i <- 1 to N)
13       yield Uniform(l,u).sample(K)
14   val y = for (i <- 1 to N) yield {
15     val phi = for(j <- 1 to K) yield
16         w(j) * x(i)(j)
17     Gaussian(phi.sum + b,tau).sample()
18   }
19   observe(x, y)
20 }}}
```

(a) A LDA model in Augur. The model specifies the distribution $p(\phi, \theta, z \mid w)$.

(b) A multivariate regression in Augur. The model specifies the distribution $p(\mathbf{w}, b, \tau \mid x, y)$.

Figure 1: Example Augur programs.

## 3   System Architecture

We now describe how a model specification is transformed into CUDA code running on a GPU. Augur has two distinct compilation phases. The first phase transforms the block of code following the `bayes` keyword into our IR for probability distributions, and occurs when `scalac` is invoked. The second phase happens at runtime, when a method is invoked on the model. At that point, the IR is transformed, analyzed, and optimized, and then CUDA code is emitted, compiled, and executed.

Due to these two phases, our system is composed of two distinct components that communicate through the IR: the front end, where the DSL is converted into the IR, and the back end, where the IR is compiled down to the chosen inference algorithm (currently Metropolis-Hastings, Gibbs sampling, or Metropolis-Within-Gibbs). We use the Scala macro system to define the modeling language in the front end. The macro system allows us to define a set of functions (called "macros") that are executed by the Scala compiler on the code enclosed by the macro invocation. We currently focus on Bayesian networks, but other DSLs (e.g., Markov random fields) could be added without modifications to the back end. The implementation of the macros to define the Bayesian network language is conceptually uninteresting so we omit further details.

Separating the compilation into two distinct phases provides many advantages. As our language is implemented using Scala's macro system, it provides automatic syntax highlighting, method name completion, and code refactoring in any IDE which supports Scala. This improves the usability of the DSL as we require no special tools support. We also use Scala's parser, semantic analyzer (e.g., to check that variables have been defined), and type checker. Additionally we benefit from `scalac`'s optimizations such as constant folding and dead code elimination. Then, because we compile the IR to CUDA code *at run time*, we know the values of all the hyperparameters and the size of the dataset. This enables better optimization strategies, and also gives us important insights into how to extract parallelism (Section 4.2). For example, when compiling LDA, we know that the number of topics is much smaller than the number of documents and thus parallelizing over documents produces more parallelism than parallelizing over topics. This is analogous to JIT compilation in modern runtime systems where the compiler can make different decisions at runtime based upon the program state.

## 4   Generation of Data-Parallel Inference

We now explain how Augur generates data-parallel samplers by exploiting the conditional independence structure of the model. We will use the two examples from Section 2 to explain how the compiler analyzes the model and generates the inference code.

When we invoke an inference procedure on a model (e.g., by calling `model.map`), Augur compiles the IR into CUDA inference code for that model. Our aim with the IR is to make the parallelism explicit in the model and to support further analysis of the probability distributions contained within. For example, a $\prod$ indicates that each sub-term in the expression can be evaluated in parallel. Informally, our IR expressions are generated from this Backus-Naur Form (BNF) grammar:

$$P ::= p(\vec{X}) \mid p(\vec{X} \mid \vec{X}) \mid PP \mid \frac{1}{P} \mid \prod_{i}^{N} P \mid \int_{X} P \, dx \mid \{P\}_c$$

The use of a symbolic representation for the model is key to Augur's ability to scale to large networks. Indeed, as we show in the experimental study (Section 5), popular probabilistic modeling systems such as JAGS [7] or Stan [8] reify the graphical model, resulting in unreasonable memory consumption for models such as LDA. However, a consequence of our symbolic representation is that it is more difficult to discover conjugacy relationships, a point we return to later.

## 4.1 Generating data-parallel MH samplers

To use Metropolis-Hastings (MH) inference, the compiler emits code for a function $f$ that is proportional to the distribution to be sampled. This code is then linked with our library implementation of MH. The function $f$ is the product of the prior and the model likelihood and is extracted automatically from the model specification. In our regression example this function is: $f(\mathbf{x}, \mathbf{y}, \tau, b, \mathbf{w}) = p(b)p(\tau)p(\mathbf{w})p(\mathbf{x})p(\mathbf{y} \mid \mathbf{x}, b, \tau, \mathbf{w})$ which we rewrite to

$$f(\mathbf{x}, \mathbf{y}, \tau, b, \mathbf{w}) = p(b)p(\tau) \left( \prod_{k}^{K} p(w_k) \right) \left( \prod_{n}^{N} p(x_n)p(y_n \mid \mathbf{x_n} \cdot \mathbf{w} + b, \tau) \right)$$

In this form, the compiler knows that the distribution factorizes into a large number of terms that can be evaluated in parallel and then efficiently multiplied together. Each $(x, y)$ contributes to the likelihood independently (i.e., the data is i.i.d.), and each pair can be evaluated in parallel and the compiler can optimize accordingly. In practice, we work in log-space, so we perform summations. The compiler then generates the CUDA code to evaluate $f$ from the IR. This code generation step is conceptually simple and we will not explain it further.

It is interesting to note that the code scales well despite the simplicity of this parallelization: there is a large amount of parallelism because it is roughly proportional to the number of data points; uncovering the parallelism in the code does not increase the amount of computation performed; and the ratio of computation to global memory accesses is high enough to hide the memory latency.

## 4.2 Generating data-parallel Gibbs samplers

Alternatively we can generate a Gibbs sampler for conjugate models. We would prefer to generate a Gibbs sampler for LDA, as an MH sampler will have a very low acceptance ratio. To generate a Gibbs sampler, the compiler needs to figure out how to sample from each univariate conditional distribution. As an example, to draw $\theta_m$ as part of the $(\tau + 1)$th sample, the compiler needs to generate code that samples from the following distribution

$$p(\theta_m^{\tau+1} \mid w^{\tau+1}, z^{\tau+1}, \theta_1^{\tau+1}, ..., \theta_{m-1}^{\tau+1}, \theta_{m+1}^{\tau}, ..., \theta_M^{\tau}).$$

As we previously explained, our compiler uses a symbolic representation of the model: the advantage is that we can scale to large networks, but the disadvantage is that it is more challenging to uncover conjugacy and independence relationships between variables. To accomplish this, the compiler uses an algebraic rewrite system that aims to rewrite the above expression in terms of expressions it knows (i.e., the joint distribution of the model). We show a few selected rules below to give a flavor of the rewrite system. The full set of 14 rewrite rules are in the supplementary material.

(a) $\frac{P}{P} \Rightarrow 1$  (c) $\prod_{i}^{N} P(x_i) \Rightarrow \prod_{i}^{N} \{P(x_i)\}_{q(i)=T} \prod_{i}^{N} \{P(x_i)\}_{q(i)=F}$

(b) $\int P(x)\, Q \, \mathrm{d}x \Rightarrow Q \int P(x) \mathrm{d}x$  (d) $P(x \mid y) \Rightarrow \frac{P(x,y)}{\int P(x,y)\, \mathrm{d}x}$

Rule (a) states that like terms can be canceled. Rule (b) says that terms that do not depend on the variable of integration can be pulled out of the integral. Rule (c) says that we can partition a product

over $N$ terms into two products, one where a predicate $q$ is true on the indexing variable and one where it is false. Rule (d) is a combination of the product and sum rule. Currently, the rewrite system is comprised of rules we found useful in practice, and it is easy to extend the system with more rules.

Going back to our example, the compiler rewrites the desired expression into the one below:

$$\frac{1}{\mathcal{Z}} p(\theta_m^{\tau+1}) \prod_j^{N(m)} p(z_{mj}|\theta_m^{\tau+1})$$

In this form, it is clear that each $\theta_1, \ldots, \theta_m$ is independent of the others after conditioning on the other random variables. As a result, they may all be sampled in parallel.

At each step, the compiler can test for a conjugacy relation. In the above form, the compiler recognizes that the $z_{mj}$ are drawn from a categorical distribution and $\theta_m$ is drawn from a Dirichlet, and can exploit the fact that these are conjugate distributions. The posterior distribution for $\theta_m$ is `Dirichlet`$(\alpha + c_m)$ where $c_m$ is a vector whose $k$th entry is the number of $z$ of topic $k$ from document $m$. Importantly, the compiler now knows that sampling each $z$ requires a counting phase.

The case of the $\phi$ variables is more interesting. In this case, we want to sample from

$$p(\phi_k^{\tau+1}|w^{\tau+1}, z^{\tau+1}, \theta^{\tau+1}, \phi_1^{\tau+1}, ..., \phi_{k-1}^{\tau+1}, \phi_{k+1}^{\tau}, ..., \phi_K^{\tau})$$

After the applying the rewrite system to this expression, the compiler discovers that it is equal to

$$\frac{1}{\mathcal{Z}} p(\phi_k) \prod_i^M \prod_j^{N(i)} \{p(w_i|\phi_{z_{ij}})\}_{k=z_{ij}}$$

The key observation that the compiler uses to reach this conclusion is the fact that the $z$ are distributed according to a categorical distribution and are used to index into the $\phi$ array. Therefore, they partition the set of words $w$ into $K$ disjoint sets $w_1 \uplus ... \uplus w_k$, one for each topic. More concretely, the probability of words drawn from topic $k$ can be rewritten in partitioned form using rule (c) as $\prod_i^M \prod_j^{N(i)} \{p(w_{ij}|\phi_{z_{ij}})\}_{k=z_{ij}}$ as once a word's topic is fixed, the word depends upon only one of the $\phi_k$ distributions. In this form, the compiler recognizes that it should draw from `Dirichlet`$(\beta + c_k)$ where $c_k$ is the count of words assigned to topic $k$. In general, the compiler detects this pattern when it discovers that samples drawn from categorical distributions are being used to index into arrays.

Finally, the compiler turns to analyzing the $z_{ij}$. It detects that they can be sampled in parallel but it does not find a conjugacy relationship. However, it discovers that the $z_{ij}$ are drawn from discrete distributions, so the univariate distribution can be calculated exactly and sampled from. In cases where the distributions are continuous, it tries to use another approximate sampling method to sample from that variable.

One concern with such a rewrite system is that it may fail to find a conjugacy relation if the model has a complicated structure. So far we have found our rewrite system to be robust and it can find all the usual conjugacy relations for models such as LDA, GMMs or HMMs, but it suffers from the same shortcomings as implementations of BUGS when deeper mathematics are required to discover a conjugacy relation (as would be the case for instance for a non-linear regression). In the cases where a conjugacy relation cannot be found, the compiler will (like BUGS) resort to using Metropolis-Hastings and therefore exploit the inherent parallelism of the model likelihood.

Finally, note that the rewrite rules are applied deterministically and the process will always terminate with the same result. Overall, the cost of analysis is negligible compared to the sampling time for large data sets. Although the rewrite system is simple, it enables us to use a concise symbolic representation for the model and thereby scale to large networks.

### 4.3 Data-parallel Operations on Distributions

To produce efficient code, the compiler needs to uncover parallelism, but we also need a library of data-parallel operations for distributions. For instance, in LDA, there are two steps where we sample from many Dirichlet distributions in parallel. When drawing the per document topic distributions, each thread can draw a $\theta_i$ by generating $K$ Gamma variates and normalizing them [9]. Since the

number of documents is usually very large, this produces enough parallelism to make full use of the GPU's cores. However, this may not produce sufficient parallelism when drawing the $\phi_k$, because the number of topics is usually small compared to the number of cores. Consequently, we use a different procedure which exposes more parallelism (the algorithm is given in the supplementary material). To generate $K$ Dirichlet variates over $V$ categories with concentration parameters $\alpha_{11}, \ldots, \alpha_{KV}$, we first generate a matrix $A$ where $A_{ij} \sim \texttt{Gamma}(\alpha_{ij})$ and then normalize each row of this matrix. To sample the $\theta_i$, we could launch a thread per row. However, as the number of columns is much larger than the number of rows, we launch a thread to generate the gamma variates for each column, and then separately compute a normalizing constant for each row by multiplying the matrix with a vector of ones using CUBLAS. This is an instance where the two-stage compilation procedure (Section 3) is useful, because the compiler is able to use information about the relative sizes of $K$ and $V$ to decide that the complex scheme will be more efficient than the simple scheme.

This sort of optimization is not unique to the Dirichlet distribution. For example, when sampling a large number of multivariate normals by applying a linear transformation to a vector of normal samples, the strategy for extracting parallelism may change based on the number of samples to generate, the dimension of the multinormal, and the number of GPU cores. We found that issues like these were crucial to generating high-performance data-parallel samplers.

### 4.4 Parallelism & Inference Tradeoffs

It is difficult to give a cost model for Augur programs. Traditional approaches are not necessarily appropriate for probabilistic inference because there are tradeoffs between faster sampling times and convergence which are not easy to characterize. In particular, different inference methods may affect the amount of parallelism that can be exploited in a model. For example, in the case of multivariate regression, we can use the Metropolis-Hastings sampler presented above, which lets us sample from all the weights in parallel. However, we may be better off generating a Metropolis-Within-Gibbs sampler where the weights are sampled one at a time. This reduces the amount of exploitable parallelism, but it may converge faster, and there may still be enough parallelism in each calculation of the Hastings ratio by evaluating the likelihood in parallel.

Many of the optimizations in the literature that improve the mixing time of a Gibbs sampler, such as blocking or collapsing, reduce the available parallelism by introducing dependencies between previously independent variables. In a system like Augur it is not always beneficial to eliminate variables (e.g., by collapsing) if it introduces more dependencies for the remaining variables. Currently Augur cannot generate a blocked or collapsed sampler, but there is interesting work on automatically blocking or collapsing variables [10] that we wish to investigate in the future. Our experimental results on LDA demonstrate this tradeoff between mixing and runtime. There we show that while a collapsed Gibbs sampler converges more quickly in terms of the number of samples compared to an uncollapsed sampler, the uncollapsed sampler converges more quickly in terms of runtime. This is due to the uncollapsed sampler having much more available parallelism. We hope that as more options and inference strategies are added to Augur, users will be able to experiment further with the tradeoffs of different inference methods in a way that would be too time-consuming to do manually.

## 5 Experimental Study

We provide experimental results for the two examples presented throughout the paper and in the supplementary material for a Gaussian Mixture Model (GMM). More detailed information on the experiments can be found in the supplementary material.

To test multivariate regression and the GMM, we compare Augur's performance to those of two popular languages for statistical modeling, JAGS [7] and Stan [8]. JAGS is an implementation of BUGS, and performs inference using Gibbs sampling, adaptive MH, and slice sampling. Stan uses a No-U-Turn sampler, a variant of Hamiltonian Monte Carlo. For the regression, we configured Augur to use MH[1], while for the GMM Augur generated a Gibbs sampler. In our LDA experiments we also compare Augur to a handwritten CUDA implementation of a Gibbs sampler, and also to

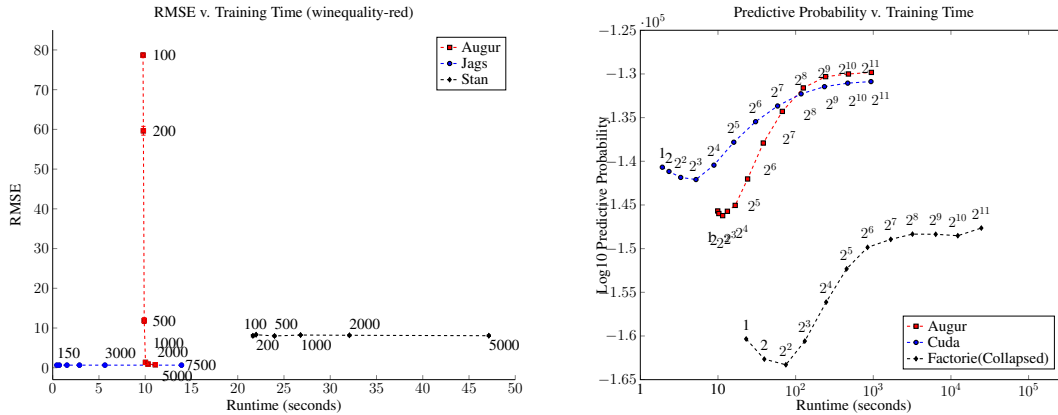

(a) Multivariate linear regression results on the UCI WineQuality-red dataset.

(b) Predictive probability vs time for up to 2048 samples with three LDA implementations: Augur, hand-written CUDA, Factorie's Collapsed Gibbs.

Figure 2: Experimental results on multivariate linear regression and LDA.

the collapsed Gibbs sampler [12] from the Factorie library [13]. The former is a comparison for an optimised GPU implementation, while the latter is a baseline for a CPU Scala implementation.

## 5.1 Experimental Setup

For the linear regression experiment, we used data sets from the UCI regression repository [14]. The Gaussian Mixture Model experiments used two synthetic data sets, one generated from 3 clusters, the other from 4 clusters. For the LDA benchmark, we used a corpus extracted from the simple English variant of Wikipedia, with standard stopwords removed. This corpus has 48556 documents, a vocabulary size of 37276 words, and approximately 3.3 million tokens. From that we sampled 1000 documents to use as a test set, removing words which appear only in the test set. To evaluate the model we measure the log predictive probability [15] on the test set.

All experiments ran on a single workstation with an Intel Core i7 4820k CPU, 32 GB RAM, and an NVIDIA GeForce Titan Black. The Titan Black uses the Kepler architecture. All probability values are calculated in double precision. The CPU performance results using Factorie are calculated using a single thread, as the multi-threaded samplers are neither stable nor performant in the tested release. The GPU results use all 960 double-precision ALU cores available in the Titan Black. The Titan Black has 2880 single-precision ALU cores, but single precision resulted in poor quality inference results, though the speed was greatly improved.

## 5.2 Results

In general, our results show that once the problem is large enough we can amortize Augur's startup cost of model compilation to CUDA, `nvcc` compilation to a GPU binary, and copying the data to and from the GPU. This cost is approximately 9 seconds averaged across all our experiments. After this point Augur scales to larger numbers of samples in shorter runtimes than comparable systems. In this mode we are using Augur to find a likely set of parameters rather than generating a set of samples with a large *effective sample size* for posterior estimation. We have not investigated the effective sample size vs runtime tradeoff, though the MH approach we use for regression is likely to have a lower effective sample size than the HMC used in Stan.

Our linear regression experiments show that Augur's inference is similar to JAGS in runtime and performance, and better than Stan. Augur takes longer to converge as it uses MH, though once we have amortized the compilation time it draws samples very quickly. The regression datasets tend to be quite small in terms of both number of random variables and number of datapoints, so it is harder to amortize the costs of GPU execution. However, the results are very different for models where the number of inferred parameters grows with the data set. In the GMM example in the supplementary,

we show that Augur scales to larger problems than JAGS or Stan. For $100,000$ data points, Augur draws a thousand samples in 3 minutes while JAGS takes more than 21 minutes and Stan requires more than 6 hours. Each system found the correct means and variances for the clusters; our aim was to measure the scaling of runtime with problem size.

Results from the LDA experiment are presented in Figure 2b and use predictive probability to monitor convergence over time. We compute the predictive probability and record the time (in seconds) after drawing $2^i$ samples, for $i$ ranging from 0 to 11 inclusive. It takes Augur 8.1 seconds to draw its first sample for LDA. Augur's performance is very close to that of the hand-written CUDA implementation, and much faster than the Factorie collapsed Gibbs sampler. Indeed, it takes the collapsed LDA implementation 6.7 hours longer than Augur to draw 2048 samples. We note that the collapsed Gibbs sampler appears to have converged after $2^7$ samples, in approximately 27 minutes. The uncollapsed implementations converge after $2^9$ samples, in approximately 4 minutes. We also implemented LDA in JAGS and Stan but they ran into scalability issues. The Stan version of LDA (taken from the Stan user's manual[6]) uses 55 GB of RAM but failed to draw a sample in a week of computation time. We could not test JAGS as it required more than 128 GB of RAM. In comparison, Augur uses less than 1 GB of RAM for this experiment.

## 6 Related Work

Augur is similar to probabilistic modeling languages such as BUGS [16], Factorie [13], Dimple [17], Infer.net [18], and Stan [8]. This family of languages explicitly represents a probability distribution, restricting the expressiveness of the modeling language to improve performance. For example, Factorie, Dimple, and Infer.net provide languages for factor graphs enabling these systems to take advantage of specific efficient inference algorithms (e.g., Belief Propagation). Stan, while Turing complete, focuses on probabilistic models with continuous variables using a No-U-Turn sampler (recent versions also support discrete variables). In contrast, Augur focuses on Bayesian Networks, allowing a compact symbolic representation, and enabling the generation of data-parallel code.

Another family of probabilistic programming languages is characterized by their ability to express all computable generative models by reasoning over execution traces which implicitly represent probability distributions. These are typically a Turing complete language with probabilistic primitives and include Venture [19], Church [20], and Figaro [21]. Augur and the modeling languages described above are less expressive than these languages, and so describe a restricted set of probabilistic programs. However performing inference over program traces generated by a model, instead of the model support itself, makes it more difficult to generate an efficient inference algorithm.

## 7 Discussion

We show that it is possible to *automatically* generate parallel MCMC inference algorithms, and it is also possible to extract sufficient parallelism to saturate a modern GPU with thousands of cores. The choice of a Single-Instruction Multiple-Data (SIMD) architecture such as a GPU is central to the success of Augur, as it allows many parallel threads with low overhead. Creating thousands of CPU threads is less effective as each thread has too little work to amortize the overhead. GPU threads are comparatively cheap, and this allows for many small parallel tasks (like likelihood calculations for a single datapoint). Our compiler achieves this parallelization with no extra information beyond that which is normally encoded in a graphical model description and uses a symbolic representation that allows scaling to large models (particularly for latent variable models like LDA). It also makes it easy to run different inference algorithms and evaluate the tradeoffs between convergence and sampling time. The generated inference code is competitive in terms of model performance with other probabilistic modeling systems, and can sample from large problems much more quickly.

The current version of Augur runs on a single GPU, which introduces another tier into the memory hierarchy as data and samples need to be streamed to and from the GPU's memory and main memory. We do not currently support this in Augur for problems larger than GPU memory, though it is possible to analyse the generated inference code and automatically generate the data movement code [22]. This movement code can execute concurrently with the sampling process. One area we have not investigated is expanding Augur to clusters of GPUs, though this will introduce the synchronization problems others have encountered when scaling up MCMC [23].

## Footnotes

[1] Augur could not generate a Gibbs sampler for regression, as the conjugacy relation for the weights is not a simple application of conjugacy rules[11]. JAGS avoids this issue by adding specific rules for linear regression.

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
