[Supplementary Material · augur_nips_final_supplementary.pdf]

# Augur: Data-Parallel Probabilistic Modeling - Supplementary Material

**Jean-Baptiste Tristan**[1]**, Daniel Huang**[2]**, Joseph Tassarotti**[3]**,**
**Adam Pocock**[1]**, Stephen J. Green**[1]**, Guy L. Steele, Jr**[1]
[1]Oracle Labs {`jean.baptiste.tristan, adam.pocock,`
`stephen.x.green, guy.steele`}`@oracle.com`
[2]Harvard University `dehuang@fas.harvard.edu`
[3]Carnegie Mellon University `jtassaro@cs.cmu.edu`

## 1   Using an Augur Model

Figure 1 gives a Scala code sample which shows how to perform inference on an Augur model. First we allocate the parameter arrays which will contain the inferred values. Then we construct the support which encapsulates these parameters. Finally we call `LDA.model.map` to generate the MAP estimate of the parameters given the observed words. The inference algorithm is chosen at runtime, and supplied as a parameter to the `LDA.model.map` call. In this case we chose to use Gibbs sampling, and thus supplied `Infer.GIBBS`.

At test time we construct a new signature containing the inferred parameters and the test documents. Then the `LDA.model.map` method is called again, with the inferred `phi` values set as observed. To test the model, a new signature is constructed containing the test documents, and the previously inferred `phi` values. Then `LDA.model.map` is called again, but with both the phis and the words observed (by supplying `Set("phi")`). The inferred thetas for the test documents are stored in `sTest.theta`.

```scala
1  val phi = new Array[Double](k * v)
2  val thetaTrain = new Array[Double](numDocTrain * k)
3  val zTrain = new Array(numTokensTrain)
4  val sTrain = new LDA.sig(phi, thetaTrain, zTrain, wTrain)
5  LDA.model.map(Set(), (k, v, numDocTrain, docsLengthTrain), sTrain, numSamples, Infer.
       GIBBS)
6
7  val thetaTest = new Array[Double](numDocTest * k)
8  val zTest = new Array(numTokensTest)
9  val sTest = new LDA.sig(phi, thetaTest, zTest, wTest)
10 LDA.model.map(Set("phi"), (k, v, numDocTest, docsLengthTest), sTest, numSamples,
       Infer.GIBBS)
```

Figure 1: Example use of the LDA model from the main paper. Function `LDA.model.map` returns a maximum a posteriori estimation. It takes as arguments the set of variables to observe (on top of the ones declared as observed in the model specification), the hyperparameters, the initial parameters, the output parameters, the number of iterations and the inference to use. The parameters are stored in `LDA.sig`.

## 2   Examples of Model Specification

We present a few examples of model specifications in Augur, covering three important topics in machine learning: regression (2.1), clustering (2.3, 2.4, 2.6), and classification (2.2, 2.5). Our goal is to show how several popular models can be programmed in Augur. For each of these examples,

```
1  object UnivariatePolynomialRegression {
2    class sig(var w: Array[Double], var b: Double, var x: Array[Double], var y: Array[
         Double])
3    val model = bayes {
4      (N: Int, M: Int) => {
5        val w = Gaussian(0,1).sample(M)
6        val x = Uniform(0,2).sample(N)
7        val b = Gaussian(0,1).sample
8        val y = for(i <- 1 to N) {
9          val monomials = for (j <- 1 to M) yield { w(j) * pow(x(i),j) }
10         Gaussian(monomials.sum + b, 1).sample()
11       }
12       observe(x, y)
13     }
14   }
15 }
```

Figure 2: Specification of a univariate polynomial regression

we first describe the support of the model, and then sketch the generative process, relating the most complex parts of the program to their usual mathematical notation.

## 2.1 Univariate polynomial regression

Our first example model is for univariate polynomial regression (Figure 2). The model's support is composed of the array w for the weights of each mononomial, x for the domain data points and y for their image. The parameters of the model are: N, the dataset size and M, the order of the polynomial. For simplicity, this example assumes that the domain of x ranges from 0 to 2.

The generative process is: We first independently draw each of the M weights, $w_i \sim N(0, 1)$, then draw $(x, y)$ as follows:

$$x_j \sim \text{Uniform}(0, 2) \tag{1}$$

$$y_j \sim N(\sum_i^M w_i x_j^i, 1). \tag{2}$$

For simplicity, the model is presented with many "hardwired" parameters, but it is possible to parameterize the model to control the noise level, or the domain of x.

## 2.2 Multivariate logistic regression

The second model is a multivariate logistic regression model (Figure 3). This is similar in form to the multivariate linear regression presented in the main paper, except it passes the weighted sum of the features through a sigmoid function, and then drawing the label from a Bernoulli distribution parameterised by the output of the sigmoid function.

## 2.3 Categorical mixture

The third example is a categorical mixture model (Figure 4). The model's support is composed of an array z for the cluster selection, x for the data points that we draw, theta for the priors of the categorical that represents the data, and phi for the prior of the indicator variable. The parameters of the model are: N data size, K number of clusters, and V for the vocabulary size.

The generative process is: For each of the N data points we want to draw, we select a cluster z according to their distribution phi and then draw from the categorical with distribution given by theta(z).

## 2.4 Gaussian Mixture Model

The fourth example is a univariate Gaussian mixture model (Figure 5). The model's support is composed of an array z for the cluster selection, x for the data points that we draw, mu for the priors

```
1  object LogisticRegression {
2    class sig(var w: Array[Double], var b: Double, var x: Array[Double], var s: Array[
         Double], var y: Array[Int])
3    val model = bayes {
4      (K: Int, N: Int, l: Double, u: Double) => {
5        val w = Gaussian(0, 10).sample(K)
6        val b = Gaussian(0, 10).sample()
7        val x = for(i <- 1 to N) yield Uniform(l, u).sample(K)
8        val y = for (i <- 1 to N) {
9          val phi = for(j <- 1 to K) yield w(j) * x(i)(j)
10         val s = sigmoid(phi.sum + bias)
11         Bernoulli(s).sample()
12       }
13       observe(x, y)
14     }
15   }
16 }
```

Figure 3: Specification of a multivariate logisitic regression

```
1  object CategoricalMixture {
2    class sig(var z: Array[Int], var x: Array[Int], var theta: Array[Double], var phi:
         Array[Double])
3    val model = bayes {
4      (N: Int, K: Int, V: Int) => {
5        val alpha = vector(V,0.5)
6        val beta = vector(K,0.5)
7        val theta = Dirichlet(V,alpha).sample(K)
8        val phi = Dirichlet(K,beta).sample()
9        val x = for(i <- 1 to N) {
10         val z = Categorical(K, phi).sample()
11         Categorical(N,theta(z)).sample()
12       }
13       observe(x)
14     }
15   }
16 }
```

Figure 4: Specification of a categorical mixture model

```
1  object GaussianMixture {
2    class sig(var z: Array[Int], var x: Array[Double], var mu: Array[Double], var sigma
          : Array[Double], var phi: Array[Double])
3    val model = bayes {
4      (N: Int, K: Int, V: Int) => {
5        val alpha = vector(V,0.1)
6        val phi = Dirichlet(V,alpha).sample()
7        val mu = Gaussian(0,10).sample(K)
8        val sigma = InverseGamma(1,1).sample(K)
9        val x = for(i <- 1 to N) {
10          val z = Categorical(K, phi).sample()
11          Gaussian(mu(z), sigma(z)).sample()
12        }
13        observe(x)
14      }
15    }
16  }
```

Figure 5: Specification of a Gaussian mixture model

```
1  object NaiveBayesClassifier {
2    class sig(var c: Array[Int], var f: Array[Int], var pC: Double, var pFgivenC: Array
          [Double])
3    val model = bayes {
4      (N: Int, K: Int) => {
5        val pC = Beta(0.5,0.5).sample()
6        val c = Bernoulli(pC).sample(N)
7        val pFgivenC = Beta(0.5,0.5).sample(K*2)
8        val f = for(i <- 1 to N) {
9          for(j <- 1 to K) {
10            Bernoulli(pFgivenC(j * 2 + c(i))).sample()
11          }
12        }
13        observe(f, c)
14      }
15    }
16  }
```

Figure 6: Specification of a naive Bayes classifier

over the cluster means, `sigma` for the priors of the cluster variances, and `phi` for the prior of the indicator variable. The parameters of the model are: `N` data size, `K` number of clusters.

The generative process is: For each of the `N` data points we want to draw, we select a cluster `z` according to their distribution `pi` and then draw from the Gaussian with mean `mu(z)` and standard deviation `sigma(z)`.

## 2.5   Naive bayes classifier

The fifth example is a binary naive Bayes classifier (Figure 6). The support is composed of an array `c` for the class and an array `f` for the features, `pC` the prior on the positive class, and `pFgivenC` an array for the probability of each binary feature given the class. The hyperparameters of the model are: `N` the number of data points, `K` the number of features and. The features form a 2-dimensional matrix but again the user has to "flatten" the matrix into an array.

The generative process is: First we draw the probability of an event being in one class or the other as `pC`. We use `pC` has the parameter to decide for each event in which class it falls (`c`). Then, for each feature, we draw the probability of the feature occurring, `pFgivenC`, depending on whether the event is in the class or not. Finally, we draw the features `f` for each event.

## 2.6   Hidden Markov Model

The sixth example is a hidden Markov model (Figure 7) where the observation are the result of coin flips. The support is composed of the result of the coin flips `flips`, the priors for each of the coins

```
1  object HiddenMarkovModel {
2    class sig(var flips: Array[Int], var bias: Array[Double], var transitionMat: Array[
         Double], var MCStates: Array[Int])
3    val model = bayes {
4      (N: Int, numStates: Int) => {
5        val v = vector(numStates,0.1)
6        val transitionMat = Dirichlet(numStates,v).sample(numStates)
7        val bias = Beta(1.0,1.0).sample(numStates)
8        val MCStates: IndexedSeq[Int] = for (i <- 1 to N) yield Categorical(numStates,
           transitionMat(MCStates(max(0, i-1)))).sample()
9        val flips = for (i <- 1 to N) Bernoulli(bias(MCStates(i))).sample()
10       observe(flips)
11     }
12   }
13 }
```

Figure 7: Specification of a Hidden Markov Model

`bias`, the transition matrix to decide how to change coin `transitionMat`, and the states of the Markov chain that indicates which coin is being used for the flip `MCStates`. The two parameters of the model are the size of the data `N`, and the number of coins being used `numStates`.

The generative process is: draw a transition matrix for the Markov chain, a bias for each of the coins, decide what coin is to be used in each state using the transition matrix, and then flip the correct coin for each state.

# 3 Rewrite Rules for the compiler

We now provide the full set of rewrite rules used by the compiler. These fall into two categories: rules which simplify the representation, and rules which rearrange the terms.

## 3.1 Simplification rules

$$1 \cdot c = c \tag{3}$$

$$\frac{c}{1} = c \tag{4}$$

$$\prod_{i=m}^{n} 1 = 1 \tag{5}$$

$$1_{true}(P) = P \tag{6}$$

$$1_{false}(P) = 1 \tag{7}$$

$$\frac{a \cdot c}{b \cdot c} = \frac{a}{b} \tag{8}$$

$$\int P(X)\mathrm{d}x = 1 \tag{9}$$

Associativity and commutativity are provided by the normalization steps.

| **Algorithm 1** Drawing M samples from a Dirichlet | **Algorithm 2** Sampling from K Dirichlets |
|---|---|

**Algorithm 1** Drawing M samples from a Dirichlet

**Input:** array $\alpha$ of size $n$
**for** M documents in parallel **do**
    **for** $i = 0$ **to** $n - 1$ **do**
        $v[i] \sim \texttt{Gamma}(a[i])$
    **end for**
    $s = \sum\limits_{0}^{n-1} a[i]$ in parallel
    **for** $i = 0$ **to** $n - 1$ in parallel **do**
        $v[i] = \frac{v[i]}{s}$
    **end for**
**end for**
**Output:** array $v$

**Algorithm 2** Sampling from K Dirichlets

**Input:** matrix $a$ of size $k$ by $n$
**for** $i = 0$ **to** $n - 1$ in parallel **do**
    **for** $j = 0$ **to** $k - 1$ **do**
        $v[i, j] \sim \texttt{Gamma}(a[i, j])$
    **end for**
    $v \times \vec{1}$
**end for**
**Output:** matrix $v$

Figure 8: Algorithms for sampling from Dirichlet distributions.

## 3.2 Rearrangement rules

$$P(X|Y) \to \frac{P(X, Y)}{\int P(X, Y)\mathrm{d}y} \tag{10}$$

$$(\prod_{i=m}^{n} P_i) \cdot (\prod_{j=m}^{n} Q_j) \to \prod_{i=m}^{n} P_i \cdot Q_i \tag{11}$$

$$\prod_{i=m}^{n} P_i \cdot Q_i \to (\prod_{i=m}^{n} P_i) \cdot (\prod_{j=m}^{n} Q_j) \tag{12}$$

$$(\prod_{i=m}^{n} P_i) \to (\prod_{i=m}^{n})1_{cond}(P_i) \cdot 1_{\neg cond}(P_i) \tag{13}$$

$$(\prod_{i=m}^{n} P_i) \to P_i \quad \text{for arbitrary } i \in \{m, \dots, n\} \tag{14}$$

$$P_i \quad \text{for arbitary } i \in \{m, \dots, n\} \to (\prod_{i=m}^{n} P_i) \tag{15}$$

$$\int P \cdot Q(X)\mathrm{d}x \to P \cdot \int Q(x)\mathrm{d}x \tag{16}$$

The condition is determined heuristically, e.g., partitioning across indicator variables. These rules are applied in a pre-defined pipeline, so they execute in a determinisitc order and the resulting rewrite system always terminates. We leave a more complex rewrite system to future work, though we have not found it necessary for any of the models described in this paper. If it fails to find a conjugacy relation, then it falls back to sampling that variable using a Metropolis-Hastings step.

## 4 Data-Parallel Sampling from Dirichlet Distributions

Algorithm 1 presents a simple way to draw from a number of Dirichlet distributions in parallel on a GPU. It works well if the number $M$ is very large. On the contrary, it is a bottleneck if $M$ is small or much lesser than the dimension of the Dirichlet distributions.

Algorithm 2 is the more complex algorithm described in the paper. This algorithm is more efficient when the number of Dirichlets is small compared to the dimension of each individual Dirichlet.

```
1  model {
2    for( i in 1:N ) {
3      y[i] ~ dnorm( y.hat[i] , 1/tau )
4      y.hat[i] <- b0 + inprod( b[1:nPred] , x[i,1:nPred] )
5    }
6    tau ~ dgamma( 1 , 1 )
7    b0 ~ dnorm( 0 , 0.01 )
8    for ( j in 1:nPred ) {
9      b[j] ~ dnorm( 0 , 0.01 )
10   }
11 }
```

Figure 9: Multivariate Regression in Jags

```
1  data {
2    int<lower=0> nPred;
3    int<lower=0> nData;
4    real y[nData];
5    matrix[nData,nPred] x;
6    vector[nData] b0vec;
7  }
8  parameters {
9    real b0;
10   vector[nPred] b;
11   real<lower=0> tau;
12 }
13 transformed parameters {
14   vector[nData] mu;
15   vector[nData] offset;
16   offset <- b0vec * b0;
17   mu <- x * b + offset;
18 }
19 model {
20   b0 ~ normal(0,10);
21   tau ~ gamma(1,1);
22   for (d in 1:nPred)
23     b[d] ~ normal(0,10);
24   y ~ normal(mu,1/sqrt(tau));
25 }
```

Figure 10: Multivariate Regression in Stan

## 5 Experimental study

This section contains additional experimental results for the multivariate linear regression, Gaussian Mixture Model and latent Dirichlet allocation models. It also provides a detailed experimental protocol for each of the models.

### 5.1 Multivariate Regression

In our regression experiment, we compare Augur against two other models, one implemented in Jags (9) and one in Stan (10). These models are both based upon the BMLR code developed by Kruschke [1]. Each system uses the same priors and hyperparameters.

The regression experimental protocol was as follows: each dataset had 10 90%/10% train/test splits generated, and each dataset was tested using 10 different random initialisations across each of the train/test splits. Then the number of samples was varied between 100, 200, 500, 1000, 2000, 5000[1]. This gives a total of 600 runs of each system on each dataset. The presented figures average across both the random seeds and the train/test splits to produce one point per number of samples. We then plot average RMSE on the test sets against average runtime.

(a) Multivariate linear regression on the Concrete Compressive Strength data set.

(b) Multivariate linear regression on the Yacht Hydrodynamics data set.

Figure 11: Multivariate linear regression experiments (I).

(a) Multivariate linear regression on the winequality-red data set.

(b) Multivariate linear regression on the winequality-white data set.

Figure 12: Multivariate linear regression experiments (II).

In Figures 11a, 12a, 12b and 11b we present results on the Concrete compressive, winequality-red, winequality-white and Yacht Hydrodynamics datasets from the UCI repository [2]. JAGS is using a Gibbs sampler for the weights and the bias, and uses a slice sampler for the variance of the noise. Augur uses random walk Metropolis-Hastings, and Stan is using the No-U-Turn variant of Hamiltonian Monte Carlo. We can see that Augur has a startup cost of about 10 seconds, and Stan has a startup cost of about 20 seconds. After that point Augur can draw samples more quickly than both Stan and JAGS, though due to JAGS's low startup time ($\ll$ 1 second) it is only on large datasets with many samples that Augur provides a speedup. Each of the algorithms converges quickly in terms of runtime, though the MH procedure used by Augur takes many more samples before reaching a satisfactory RMSE. We suspect that the Stan samples provide a better characterisation of the posterior than the samples from JAGS or Augur, as HMC and its variants are better at exploring a distribution. It would be interesting to use a GPU-based automatic differentiation system and integrate this into Augur, as Augur already provides fast parallel likelihood evaluations, and so it should be possible to generate a high quality HMC sampler on the GPU.

The RMSEs of JAGS and Augur converge to approximately similar values, though Augur takes longer to converge (in terms of the number of samples, and total runtime) as Metropolis-Hastings is a less efficient inference algorithm for regression than a tuned Gibbs sampler. As mentioned in

```
1  model {
2    for (i in 1:N){
3      z[i] ~ dcat(theta)
4      y[i] ~ dnorm(mu[z[i]],sigma[z[i]])
5    }
6    theta[1:K] ~ ddirch(alpha)
7    for (k in 1:K) {
8      alpha[k] <- 1
9      mu[k] ~ dnorm(0,0.01)
10     sigma[k] ~ dgamma(1,1)
11   }
12 }
```

Figure 13: GMM in Jags

section 5 of the paper JAGS has a special case for working with linear regression models which alters the sampling procedure, and this feature is not currently available in Augur.

We find that the regression results show that Augur is competitive with other systems, though the linear regression datasets we tested are not large enough to properly exploit all the computation available in the GPU.

## 5.2 Gaussian Mixture Model

The Gaussian Mixture Model results described in Section 5 of the paper show how each of the three systems scale as the dataset size is increased. We sampled $100,000$ datapoints from two different mixture distributions: one with 4 gaussians centered at {-5,-1,1,5} with standard deviation {1,0.1,2,1}, and one with 3 gaussians centered at {-5, 0, 5} with standard deviations {0.1,0.1,0.1}. Each dataset had a flat mixing distribution, that is draws from each gaussian were equiprobable. From each dataset we subsampled smaller datasets using 100, 1000 and $10,000$ datapoints.

We used the GMM presented in the paper for Augur, for Stan we used the GMM listed in the modeling handbook, and for JAGS we wrote a standard GMM (shown in figure 13), based upon Augur's. Each model used the same prior distributions and hyperparameters.

Figure 14 presents the runtimes of the two datasets, Figure 14a has 4 centres, and Figure 14b has 3 centres. For computational reasons we stopped Stan's final run after 3 hours on the second dataset (Stan took approx. 6 hours to complete on the first dataset). Here we can see that Augur's runtime scales much more slowly as the dataset size is increased. JAGS remains reasonably competitive until $100,000$ data points, at which point Augur is faster by a factor of 7. Stan is also relatively competitive but scales extremely poorly as the number of datapoints is increased.

## 5.3 LDA

In an attempt to confirm the result presented in the paper, we present another result (figure 15a) measuring the predictive probability averaging across multiple runs using different train/test splits. In this experiment, we averaged across 10 runs with different train/test splits and present the timings with error bars. We also ran an experiment across 10 different random initializations and seeds, and all algorithms again showed robustness to the variation. We reduced the maximum number of samples to 512 as generating results for the Collapsed Gibbs sampler was proving prohibitive in terms of runtime for repeated experiments.

A third experiment (Figure 15b) gives on the natural logarithm of run time in milliseconds to draw 512 samples as the number of topics varies. Augur's Gibbs sampler is linear in the number of topics during the step of sampling each of the $z_{ij}$. The collapsed Gibbs sampler's runtime increases greatly as the number of topics is increased. Again, Augur's generated code is on par with the hand-written CUDA implementation.

We experimented with the SparseLDA [3] implementation which is Factorie's default LDA model, but this implementation proved to be unreliable. The predictive probability measure actually decreased as more samples were drawn using the SparseLDA implementation. We are working with the developers of Factorie to investigate this problem. The SparseLDA implementation is interesting

(a) Runtime to draw a thousand samples from a GMM for varying data set sizes, with 4 centres. Stan's last data point is cropped, it took 380 minutes.

(b) Runtime to draw a thousand samples from a GMM for varying data set sizes, with 3 centres. Stan's 100,000 data point was not generated.

Figure 14: Gaussian Mixture Model runtimes.

(a) Average over 10 runs of the predictive probability vs time.

(b) Comparison of the scalability of Augur, hand-written CUDA, and Factorie's collapsed Gibbs *w.r.t* the number of topics.

Figure 15: Further experiments with LDA.

as it uses a set of LDA specific assumptions to generate a highly optimised Gibbs sampler. We found Augur to be competitive in terms of runtime when drawing more than 256 samples. With smaller sample sizes there is insufficient computation to amortize the compilation costs.

## Footnotes

[1]Stan and JAGS had a burn in of an additional 50% samples to allow for the adaptive tuning of the samplers, without these extra samples for adaptation the performance of both of them was poor. Augur's Metropolis-Hastings algorithm does not use such tuning.