[Reviews · NeurIPS 2014]

Submitted by Assigned_Reviewer_24

The contribution of this paper is probabilistic programming language that supports parallel inference for graphical models (specifically Bayes nets). Probabilistic programming languages are powerful tools because they allow rapid development of new models without having to derive/implement new inference algorithms. Unlike most existing probabilistic programming languages, Augur produces massively parallel code that can run on a GPU (using CUDA).

A unique feature of Augur is that it compiles the model (specified in the language Scala) into an intermediate representation before it's ultimately compiled into a CUDA inference algorithm for parallelization. This intermediate representation is advantageous because it allows Augur's compiler to optimize the code after it sees the data (much like a query optimizer might work when determining the order to do joins/selections/projections in databases).

The paper also raises important questions about parallelism and inference tradeoffs (Section 4.4). For example, more sophisticated machine learning algorithms such as collapsed samplers often have better convergence rates than the uncollapsed versions. However, collapsing introduces more dependencies which might make parallelism more difficult. In the case of LDA the parallel version of uncollapsed is substantially faster than the parallel version of collapsed (a digression: this advantage depends on having access to enough GPUs: is the compiler aware of the amount of parallelism available when choosing between the collapsed and uncollapsed sampler?). Studying these tradeoffs in the context of selecting an optimal inference algorithm could be an interesting direction of future machine learning work.

Some potential weaknesses of Augur are that it only supports directed Bayesian networks, focuses heavily on MCMC inference, and seems to only support GPU parallelism. I cant' help but wonder if choosing GPUs as the compilation target might limit its adoptability (are there many research groups that have access to large GPU clusters?).

Overall the authors do an adequate job positioning their work in the space of probababilistic programming languages, but the work could be strengthened with a more in depth comparison by (1) providing more detailed comparisons of Augur with other languages (e.g., is Augur the only language that has an intermediate representation), and (2) providing comparison with work on parallelizing inference (for example the Smola paper mentioned below, chromatic sampling, Graph Lab, recent work from the database community on big data analytics such as Chris Re's work, OptiML, etc.).

The experiments demonstrate that Augur can compete in the space of probabilistic programming languages. Three problems are evaluated (linear regression, mixture of Gaussians, and LDA). It was especially interesting to see that the massively parallel version of the basic Gibbs sampler was significantly faster than the parallel version of collapsed Gibbs sampler implemented in Factorie (though, Smola 2010 "An Architecture for Parallel Topic Models" might provide better parallelization of the collapsed Gibbs sampler than Factorie).
Summary: Well written, easy to follow paper describing a novel probabilistic programming language (Augur) that automatically compiles modeling code into highly parallel CUDA code that can run on GPUs. In general, probabilistic programming languages (and other tools/packages) have high impact potential, especially for application domains that depend on large-scale machine learning (like NLP). Augur differs from existing probabilistic programming languages in that it uses a symbolic intermediate representation that allows its compiler to be "data-aware" and thus compile into massively data-parallel inference algorithms. Related work could be more expansive.

Submitted by Assigned_Reviewer_39

This paper introduces a new probabilistic programming language for Bayesian networks, Augur, and a compiler that are designed to produce parallel inference with GPU. The whole system is composed of two parts, the front end, which converts the model description embedded in Scala into a symbolic representation, and the back end, which generates CUDA code from the intermediate representation at runtime. Compared to other probabilistic languages, Augur is different in that it is designed to produce parallel inference method for scalability, and that it uses a compact symbolic representation for distributions.

It is a meaningful contribution to automatically produce a parallel inference program for a general Bayesian model. Also, the LDA experiments addresses a common scalability problem in probabilistic programming that much more memory would be used in a probabilistic program than a hand-written program. I find it not difficult to understand the main idea of this paper, although a lot of details such as the construction of IR and the inference configuration in Augur are omitted.

My main concern is about the efficiency and the overhead of the automatically generated CUDA code:
First, it is not trivial to write a CUDA program to achieve high parallelization given a problem, and the efficiency depends highly on the number of independent tasks and the memory usage. The current proposed method seems to simply take a product expression in the symbolic representation and parallelize the computation on GPU. Is there a general mechanism for the compiler to detect which product should be parallelized and which should be rather run sequentially on CPU?
Second, the authors discusses two algorithms to optimize the sampling of multiple Dirichlet distributions, and the compiler will take the better one according to runtime values. Does this strategy require extra work to apply to a new distribution, or can you make it a general rule for all similar models?
Lastly, AFAIU the CUDA code is generated once at the beginning of a MAP/Sampling call with a constant time cost, and the currently values of the model variables is used to optimize the generated code. However, values such as the number of subtasks for a product operation can depend on other variables and is changed during the execution. Would that mean a recompilation of the CUDA code?

Another concern is the limited expressive capability of Augur. The currently language does not allow a variable size for the model support. So it can not be used for many model selection problems, e.g., a Gaussian mixture model with the number of clusters being a random variable with a prior distribution. Can author show that Augur supports all the Bayesian networks?

The experiment results for the multivariate linear regression is not impressive. The parallelized code by Augur performs similarly to the sequential code by Jags. Also, the labels and plots in Figure 2(a) are unreadable. For the LDA experiments, it is well known that the collapsed sampler converges faster than the uncollapsed sampler as a function of the sample size. However, I am suprised that Figure 2(b) shows that the Factorie converges slower and also to a significantly lower value of the predictive probability than the other methods.

--------------

Updates after feedback:
The authors feedback addresses my questions on the efficiency of the CUDA code to my satisfaction. However, their response on Factorie confirms my concern on the comparison between CUDA parallelized sampler and the collapsed Gibbs sampler. The plot of Factorie does not show the actual performance of collapsed Gibbs on the predictive probability. It is therefore hard to judge if Augur achieves the same performance as collapsed Gibbs given the same amount of time. I suggest the authors discusses this issue explicitly in a final version of the paper.
Summary: This paper introduces a new probabilistic programming language for Bayesian networks and a compiler that are designed to produce parallel inference with GPU. It achieves scalability but has limited modelling power.

Submitted by Assigned_Reviewer_41

This paper proposes a Scala-based language for BUGS-style probabilistic programming that automatically discovers parallelism and exploits it using generated GPU code. I am not an expert in programming languages and compilers, but the basic scheme seems reasonable and effective. Certainly having probabilistic programming languages that can compile to GPUs would be a very welcome contribution!

The experimental results would have been somewhat more compelling if they had included an evaluation of *effective* samples per second as well as raw samples per second to correct for autocorrelation in the Markov chain. Looking at held-out metric (though important) is not the same thing; both Gibbs and M-H may get to a small region of parameter space with high posterior probability and then get stuck there. This is not “convergence” in the traditional sense of a Markov chain reaching its stationary distribution.

Certainly there are plenty of machine learning applications (rather than Statistics applications) where MCMC is used as a stochastic search algorithm, and where high-quality uncertainty estimates are of little or no interest. (LDA is a prime example.) In these cases Gibbs or Metropolis may be better choices than NUTS. But it would be nice to know how useful Augur is for statistical applications where (for example) having good error bars around your regression coefficients is desirable.

That said, the experiments support the conclusion that Augur finds useful parameter settings faster than the competition, which is in itself a very useful contribution.

Some minor comments:

Line 63: LDA should be spelled out.

Figure 1: Why is the support called “sig”?

Line 229: This notation is a little sloppy (which wouldn’t be a big deal, except that we’re talking about compiled code). theta^{tau+1}_m should have different names in the numerator and denominator.

Section 4.3: How efficient are the standard random number generation libraries for GPUs? My impression was that standard techniques like rejection sampling are much less efficient on GPUs, making it difficult to sample from (for example) gamma distributions efficiently.

Figure 2a is very difficult to read, since the numbers get in the way of the (very small) ticks that say which sampler is which.

Figure 15: The legend is broken.
Summary: This paper is well written and presents a data-parallel probabilistic programming language that can exploit modern GPU hardware. The experimental evaluation has one very serious flaw, which needs to be corrected—other than that the paper seems pretty strong.
Author Feedback
Author rebuttal: Comments common to all reviewers:

We will revise the experimental figures to improve readability (and fix the broken legend in Figure 15).

Reviewer 1 (assigned ID 24):

We agree it would be interesting to compare to Smola 2010, but we could not find an implementation with a compatible license for all the dependencies. Augur's rewrite system is not currently complex enough to generate a collapsed sampler (and they are inherently sequential).

Reviewer 2 (assigned ID 39):

Our compiler does not always parallelize a product, it makes a decision about the amount of work available in each parallel task before deciding which products and sampling steps to parallelize. This decision is made at runtime based upon the size of the parameters. E.g., in LDA corpora with many small documents the compiler parallelizes across documents, but does not parallelize across words in a document. In corpora with a few very large documents it parallelizes across both words and documents. In some cases it uses a single thread to sample part of the model and in other cases it parallelizes along one of the dimensions. E.g., the Dirichlet in the GMM is sampled using a single thread for small numbers of clusters. All computation is either all GPU or all CPU, as the memory transfer overhead is prohibitive.

Currently the only distribution with two sampling procedures is the Dirichlet, but the general principle of selecting the parallelization dimension to maximise the parallel work is part of the compiler. To extend it to new distributions would require some work to implement the transposed sampler for the specific new distribution, but then the compiler would choose from the available parallelization options without further extension.

As you stated the CUDA code is only compiled once, and the current values of the hyperparameters and the data control the parallelization. We currently do not support models with variable size support (after the submission deadline we finished preliminary work on Dirichlet Processes, thus allowing the support to change size, though the details are too long for this response).

In our LDA results we look at the runtime until convergence rather than the number of samples (as our experimental results show the collapsed sampler converges in fewer samples, but a longer overall runtime).

The Factorie implementation has some smoothing applied when the topics are generated from the collapsed sampler which makes it difficult to directly compare the predictive probability values. Also we refer the reviewer to "On smoothing and inference for topic models" Asuncion et al 2009 to note the impact that hyperparameter tuning has on different inference methods, and regret we did not have time to perform this tuning in our results.

Reviewer 3 (assigned ID 41):

We completely agree with the statements about effective sample size, and we are happy to revise our experimental section to clarify that we are comparing the time to find a parameter setting, and we do not consider the effective sample size for posterior distribution estimation. We agree that it is very likely that a NUTS sampler will provide a larger effective sample size than a MH sampler, and will revise the text to make this clear to the reader.

Sig is short for signature, and it probably should be renamed support. The compiler expects to find a member variable called sig, which it uses as the support. It would be simple to change the compiler to rename it support.

We haven't found RNG to be a bottleneck in our experiments, and the CUDA generators for uniform and gaussian numbers are fairly good. We implemented the other forms of RNG ourselves, based upon techniques from the literature. The probability of the gamma RNG we used going through more than 2 rounds of rejection sampling is very small, and as a consequence we found this did not cause much divergence in a warp.